# The pseudo-global-warming (PGW) approach: Methodology, software package PGW4ERA5 v1.1, validation and sensitivity analyses

Roman Brogli[1,2], Christoph Heim[1], Jonas Mensch[1], Silje Lund Sørland[1,3], and Christoph Schär[1]

[1]Institute for Atmospheric and Climate Science, ETH Zurich, Universitätstrasse 16, 8092 Zurich, Switzerland
[2]SRF Meteo, 8052 Zurich, Switzerland
[3]NORCE, Jahnebakken 5, 5007 Bergen, Norway

**Correspondence:** Christoph Heim (christoph.heim@env.ethz.ch)

**Abstract.** The term pseudo-global warming (PGW) refers to a simulation strategy in regional climate modeling. The strategy consists of directly imposing large-scale changes in the climate system on a control regional climate simulation (usually representing current conditions) by modifying the boundary conditions. This differs from the traditional dynamic downscaling technique where output from a global climate model (GCM) is used to drive regional climate models (RCM). The PGW climate changes are usually derived from a transient global climate model (GCM) simulation. The PGW approach offers several benefits such as lowering computational requirements, flexibility in the simulation design, and avoiding biases from global climate models. Yet, implementing a PGW simulation is non-trivial and care must be taken not to deteriorate the physics of the regional climate model when modifying the boundary conditions. To simplify the preparation of PGW simulations, we present a detailed description of the methodology and provide the companion software PGW4ERA5 facilitating the preparation of PGW simulations. In describing the methodology, particular attention is devoted to the adjustment of the pressure and geopotential fields. Such an adjustment is required when ensuring consistency between thermodynamical (temperature and humidity) changes on the one hand, and dynamical changes on the other hand. It is demonstrated that this adjustment is important in the extratropics, and highly essential in tropical and sub-tropical regions. We show that climate projections of PGW simulations prepared using the presented methodology are closely comparable to traditional dynamic downscaling for most climatological variables.

## 1   Introduction

Climate simulations are an essential tool to study the expected response of the climate system to greenhouse gas emissions (Flato et al., 2013). Global coupled climate models (GCMs) are used to study the earth's entire climate system, while regional climate models (RCMs) provide a more detailed description on regional to local scales. These higher resolution data are crucial to assess the impact of climate change and to establish regional mitigation and adaptation strategies (Giorgi et al., 2008; Rummukainen, 2010; Sørland et al., 2020). RCMs are forced by results from GCMs and operate on a finer grid to provide increased detail, a technique that is known as dynamic downscaling (Flato et al., 2013). The standard approach to investigate

climate change using RCMs consists of downscaling two time slices from a GCM simulation, one for the future and one for the past, and comparing them, or alternatively downscale a centennial transient simulation (see e.g. PRUDENCE (Christensen and Christensen, 2007), ENSEMBLES (der Linden and Mitchell, 2009) and CORDEX (Kotlarski et al., 2014)).

While the standard downscaling approach is indispensable for regional climate research, it has several limitations and challenges. For instance, coordinated downscaling efforts require a lot of computing power and data storage. An evaluation run, where the RCM is downscaling a global reanalysis, is required to assess the model performance. Additionally, transient climate simulations are needed to assess the regional climate change. This should be done for multiple RCMs, GCMs, and different emission scenarios, to properly represent the full uncertainty range (Hawkins and Sutton, 2009). For several research groups, such applications are beyond the limit of their computational infrastructure (Prein et al., 2015). Another limitation with the traditional downscaling approach is the uncertainty associated with the atmospheric circulation from the GCMs. Even though the RCMs reduce some of the biases from the GCMs (Sørland et al., 2018), the RCMs will not be able to correct for biases in the large-scale circulation from GCMs (Hall, 2014).

## 1.1 Concept of the PGW approach

In recent years, pseudo-global warming (PGW) simulations, have been increasingly used in research as an alternative regional climate modeling strategy. In a PGW simulation, we directly impose selected changes in the climate system on a historical regional climate simulation by modifying the initial and boundary conditions (Schär et al., 1996; Wu and Lynch, 2000; Sato et al., 2007; Rasmussen et al., 2011; Liu et al., 2017; Adachi and Tomita, 2020). In simple mathematical terms, the pseudo-global warming concept can be expressed as

$$PGW = CTRL + \Delta, \tag{1}$$

where $CTRL$ and $PGW$ represent the boundary conditions of two RCM simulations of the past and future climate, respectively, and $\Delta$ are the future changes often referred to as climate change deltas. $\Delta$ must be computed from a separate climate projection as

$$\Delta = SCEN - HIST, \tag{2}$$

where $SCEN$ is a future time slice of a climate projection and $HIST$ is the corresponding historical time slice coming from a GCM or RCM simulation. Both $SCEN$ and $HIST$ periods must be chosen long enough to reduce the effects of internal variability (average of ~30 years). While the general concept as shown by Equation (1) is common to most PGW simulations in the literature, the design of both $\Delta$ and $CTRL$ varies. In Section 2, we will describe in detail how $\Delta$ can be derived in practice. We also provide software written in Python to perform this task. Generally, $\Delta$ has to be made up of changes in temperature, humidity, wind, and pressure/geopotential. Thereby it is essential to maintain the physical balances in the perturbed boundary conditions of the $PGW$ simulation, in particular the hydrostatic balance. Violations of the hydrostatic balance may occur due to the implied temperature changes, but also due to differences in the vertical coordinate and topography between the driving GCM and the RCM boundary conditions. After applying $\Delta$ to the thermodynamic variables, it is thus essential to restore this

balance using an adequate pressure adjustment (see Section 2). Figure 1 shows an example of a PGW-driven RCM simulation along with the corresponding $CTRL$ simulation.

At first glance it is surprising that one can change the driving fields of an RCM simulation in an ad-hoc fashion as depicted above, without introducing serious inconsistencies in the atmospheric dynamics. Indeed changes in temperature $T$ will imply changes in pressure and horizontal pressure gradients, thereby affecting the hydrostatic and geostrophic balance of the atmospheric flow. If such changes are enforced inconsistently, the atmosphere will respond with a geostrophic adjustment process. Such an adjustment is potentially important, and indeed it has been the root cause behind the failure of the first numerical weather prediction forecast of Lewis F. Richardson (Lynch, 2006). However, the PGW approach rests on a solid theoretical foundation. If $\Delta T_v$ is only a function of pressure, i.e. if $\Delta T_v = \Delta T_v(p)$ where $T_v$ denotes the virtual temperature, the prescribed thermal modification does not modify the horizontal gradient of the geopotential on pressure surfaces, and hence there is no change to the dynamical forcing (Schär et al., 1996). In essence, the balance of forces is unchanged, irrespective of a temperature change $\Delta T_v(p)$. When considering the more general case of baroclinic temperature changes, i.e. when $\Delta T_v = \Delta T_v(x,y,p)$, the situation becomes more complex and a simple theoretical argument does not appear to exist. However, a number of numerical studies have demonstrated that undesirable inconsistencies do not occur or are negligibly small. Early studies of this type include those of Kimura and Kitoh (2006) and Sato et al. (2007).

## 1.2 Advantages and disadvantages of PGW simulations

Why and when could a PGW simulation be useful? We summarize potential advantages and disadvantages below:

1. The PGW approach makes climate projections with a comparatively short simulation duration possible. $\Delta$ is desinged to have a seasonal cycle, but no interannual variability. Thus, the same $\Delta$ is used to modify the boundary conditions of each simulated year. Therefore, there is no change in interannual variability in the future projection. As a result, a period shorter than the typically used 30 years suffices to assess climate change (Yoshikane et al., 2012). This is especially attractive for producing computationally demanding high-resolution climate projections (Schär et al., 2020).

2. PGW simulations can also reduce the computational and storage cost of climate projections if a reanalysis driven evaluation simulation is pre-existing. Using the PGW approach, this simulation can be modified and thus only one additional simulation is necessary to assess climate change (no need to dynamically downscale a past and future period from a GCM). The PGW approach is thus attractive when considering a multi-model ensemble of high-resolution simulations (e.g. Pichelli et al., 2021).

3. For the $CTRL$ period, PGW simulations "inherit" the synoptic environment and weather situation from the reanalysis at the lateral boundaries. Thus, the frequency of weather systems entering the domain and the large-scale synoptic situation in the $PGW$ simulation closely matches the reanalysis (Figure 1). On the one hand, this greatly reduces biases during the $CTRL$ period in comparison to conventional downscaling (where the RCM inherits biases from the driving GCM CTRL). Biases during CTRL are a considerable challenge in impact assessment. For instance a temperature bias of a few degrees implies biases in snow line of several hundred meters. The PGW methodology also allows for directly assessing

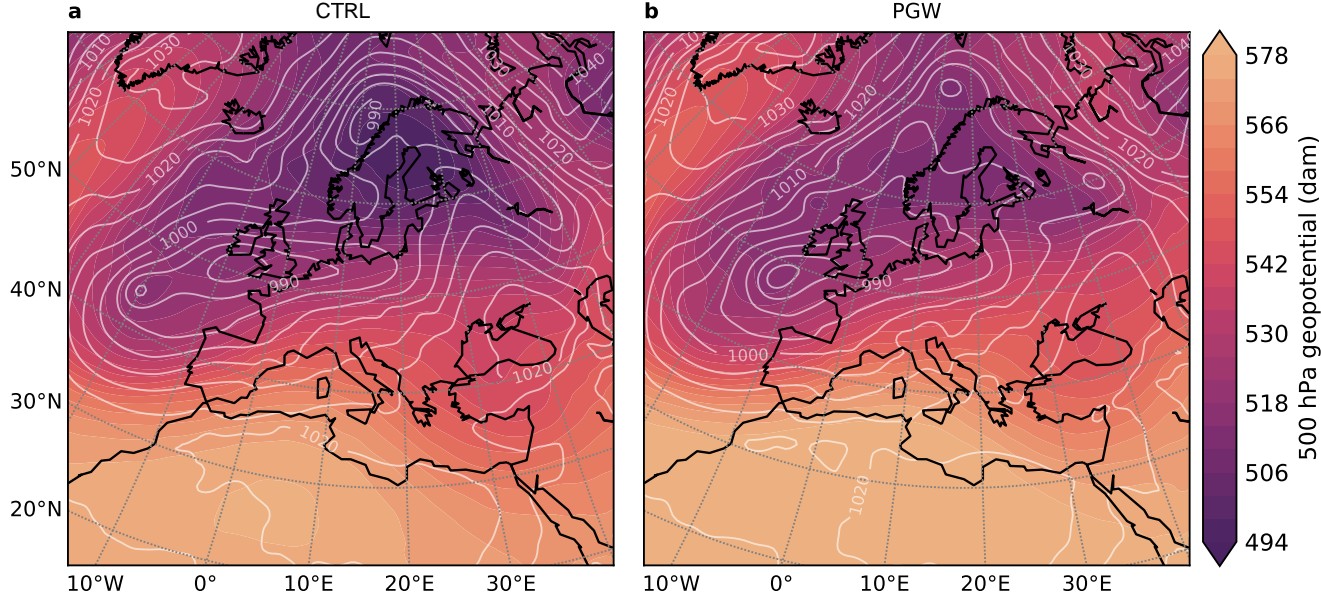

**Figure 1.** Snapshot of 500 hPa geopotential and mean sea level pressure in a $CTRL$ and $PGW$-driven simulation representing past and future conditions, respectively. Shading shows the geopotential as indicated by the colorbar. White contours show mean sea level pressure in hPa. (a) evaluation run driven by the ERA-Interim reanalaysis. (b) PGW simulation of the same timestep. The time of the evaluation run is the 30th of December 2009 at 0 UTC. More details on the model set up can be found in Table A1. Note how the lateral boundary forcing serves to approximately maintain the circulation in the lateral boundary zone, while the system evolves freely in the inner of the domain. Also, the geopotential is generally higher in the PGW simulation shown in (b) because of higher atmospheric temperatures.

how an observed historical event could look like in a different climate. This approach has recently been recommended as the "storyline" approach to vulnerability assessment (Hazeleger et al., 2015; Shepherd, 2019). On the other hand, using the same synoptic forcing in both $CTRL$ and $PGW$ implies that potential changes in intra- and interannual variability might be missed in the PGW approach.

4. For calculating $\Delta = SCEN - HIST$, only data from a few variables of the driving climate projection are needed. This means that not the full GCM data needs to be downloaded and preprocessed to run a RCM simulation. $\Delta$ can even be derived from monthly mean data.

5. $\Delta$ can not only represent input from a single GCM or RCM, but also an ensemble mean of a set of simulations can be used to drive a PGW simulation.

PGW simulations are not a standard approach included in regional climate model codes. Thus, preparing PGW simulations needs manual work to produce the initial and lateral boundary conditions. With the software described here, we try to simplify this process.

### 1.3 Applications

A common application of PGW simulations in research is to investigate a question of the type: How will certain historical or observed events change in a different climate? Such events can be tropical cyclones (Lynn et al., 2009; Ito et al., 2016; Sørland and Sorteberg, 2016; Gutmann et al., 2018; Patricola and Wehner, 2018; Jung and Lackmann, 2019), mesoscale convective systems (Prein et al., 2017; Haberlie and Ashley, 2019), atmospheric rivers (Dominguez et al., 2018), droughts (Seneviratne et al., 2002; Ullrich et al., 2018) or similar phenomena.

Since they allow for computationally cheap climate projections, PGW simulations can be used to replace standard downscaling, for example, to investigate changes in precipitation (Sato et al., 2007; Kawase et al., 2009; Taniguchi, 2016; Dai et al., 2020; Rasmussen and Liu, 2017; Wang and Wang, 2019), local temperature (Adachi et al., 2012; Expósito et al., 2015), clouds (Hentgen et al., 2019) or snow cover (Hara et al., 2008; Kawase et al., 2013; Ikeda et al., 2021).

Applications beyond the two mentioned above have also been explored. PGW simulations can be used to quantify the role of different drivers of climate change (Rowell and Jones, 2006; Kendon et al., 2010; Kröner et al., 2017; Keller et al., 2018; Brogli et al., 2019a, b).

The PGW approach can also be used to debias a GCM simulation when using it for conventional downscaling with an RCM (similar as in Misra and Kanamitsu, 2004). In this case, one has to use some reanalysis such as ERA, and define the climatological mean $\Delta$ as $\Delta = ERA - HIST$, representing the GCM bias. The debiased control simulation is then driven by $CTRL = HIST + \Delta$ ($HIST$ here represents the GCM output at full temporal resolution), and similar for the scenario simulation. This removes the mean bias of the driving GCM simulation.

### 1.4 Goal & Outline

With this article, we aim to facilitate the future generation of reanalysis-driven PGW simulations by providing the software PGW4ERA5 (Pseudo-Global Warming for ERA5) designed for this purpose along with the article. In Section 2 we describe in detail the methodology used in the software package. In Section 3, we will present the properties of PGW simulations prepared with our methodology. Even though the PGW approach is often used, such studies on the general simulation properties are sparse (e.g. Yoshikane et al., 2012). Before concluding, we present multiple sensitivity tests regarding the design of PGW simulations in Section 4. We omit an extensive review of the literature on the subject as this can be found in Adachi and Tomita (2020).

## 2 Methodology and Software

In this section we describe in detail the PGW methodology and its implementation in our Python software PGW4ERA5. We have performed multiple tests to arrive at the described strategy, many of which will be presented in Section 4.

### 2.1 Reanalysis data

The PGW4ERA5 software is designed to facilitate the derivation of the $PGW$ boundary conditions for a reanalysis-driven $CTRL$ simulation. In principal, the concepts are applicable for any kind of reanalysis, but PGW4ERA5 is designed for the use of the European Center for Medium Range Weather Forecast (ECMWF) ERA5 Re-Analysis (Hersbach et al., 2020). The standard ERA5 reanalysis output is given on a hybrid sigma-pressure coordinate. While output on pressure levels is also available and can be used to drive RCM simulations, we here focus on the use of ERA5 data on the native hybrid coordinate. Using ERA5 data on the native vertical coordinate has the advantage that the full vertical resolution is used. This is for instance essential in cases with pronounced inversions. Extending the code to the use of ERA5 data on pressure levels may be subject of future code developments and would require updating the pressure adjustment (see Section 2.8).

### 2.2 GCM data

The climate deltas used in the PGW approach should represent differences between two climatological periods. In practice we take the differences of two extended periods (e.g. 30 years) from a transient scenario simulation, representing for instance the changes between the recent climate and the end of the century for some greenhouse and aerosol emission scenario. We have been working with three different sources of CMIP GCM data obtained from the Earth System Grid Federation download portal (Cinquini et al., 2014):

1. Complete GCM output: Daily or subdaily three-dimensional data provided on the original GCM grid in terrain-following coordinates. Such data is only available for some GCMs, and usually not globally, but only over certain domains. The CMIP6 output group CFday is one example of such data that is available for a small number of models.

2. AMON data: Average monthly mean data on 19 pressure levels available from virtually all CMIP simulations. Experience with this data suggest that the vertical resolution is sufficient for applications in the extratropics.

3. EMON data: For a selection of models, a similar data set exists with a higher vertical resolution with 27 pressure level. In simulations where strong inversions are present (e.g. over the tropical and subtropical oceans), the higher resolution of the EMON data should be beneficial.

While in this paper we show some examples using full GCM output data, the presentation of the procedure does focus on the case where we use pressure-level GCM data. The GCM data used to construct the climate-change signal is listed in Tab 1. This includes the usual atmospheric 3D fields. As to be demonstrated later (see Subsection 4.3), it is advantageous to use $RH$ rather than $q_v$; if only the latter is available, $q_v$ is converted into $RH$. There are alternative approaches to treat the humidity change,

**Table 1.** Data requirements: Pressure-level monthly-mean data from the driving GCM.

| Type | Symbol | Variable |
|------|--------|----------|
| 3D | $T$ | temperature |
| 3D | $(u, v)$ | horizontal wind |
| 3D | $RH$ | relative humidity |
| 3D | $\phi$ | geopotential |
| 2D | $T_{2m}$ | 2m temperature |
| 2D | $RH_{2m}$ | 2m relative humidity |
| 2D | $SST$ | sea surface temperature |
| 2D | $T_{\text{sfc}}$ | surface skin temperature |

such as for instance, the assumption of no change of $RH$ with warming (e.g. Adachi and Tomita, 2020). Such modifications are straightforward to implement through the modification of $\Delta RH$. We note that caution is required for temperatures below 0°C where the definition of $RH$ may differ between models. Our software uses the definition of saturation vapor pressure over water and ice following the implementation in the IFS model.

The geopotential $\phi$ is also available as a 3D field. However, in the simplest case we only require one level, i.e. $\phi_{ref}$ at reference pressure level $p_{ref}$. This information is sufficient to reconstruct the three-dimensional geopotential field using the hydrostatic equation. The pressure level $p_{ref}$ must be chosen to be located above the surface throughout the geographical domain considered. In cases with high topography, one may also use different reference pressure levels in different areas of the domain (see Subsection 2.9.2).

The procedure uses 2m-data for $RH_{2m}$ and $T_{2m}$ to improve the vertical interpolation of $T$ and $RH$ near the surface. The surface and soil temperature change is derived using $T_{\text{sfc}}$ and $SST$ (see Subsection 2.7). Soil moisture data is not used as different model's soil moisture have different meanings. Initial soil moisture for the RCM simulation must thus be reconstructed by a simulation over an extended spin-up period.

The climate change delta for each of the variables $\chi$ listed above is computed on the GCM mesh as

$$\Delta\chi = \chi_{SCEN} - \chi_{HIST}. \tag{3}$$

The $SCEN - HIST$ differences represent averages of two climatological periods (e.g. 30 years each), either at monthly (Jan-Dec) or daily resolution. In PGW4ERA5, the climate deltas are stored and processed in the netCDF format (http://doi.org/10.5065/D6H70CV The handling of the netCDF-files is done with the Python package xarray (Hoyer and Hamman, 2017).

## 2.3 Overview of workflow

The PGW4ERA5 software package uses as input the underlying GCM simulation(s) and the reanalysis (abbreviated as ERA in the following). The package outputs the PGW-modulated reanalysis. The overall workflow of PGW simulations is shown in Fig. 2. The figure uses the convention that the subscripts denote the respective computational mesh (see caption). The

RCM simulation under past or current climate conditions ($CTRL_{RCM}$) follows the usual procedure of an ERA-driven RCM simulation: ERA is interpolated to the RCM grid with a temporal resolution of a few hours, and then drives the RCM simulation at the lateral and lower boundaries. The interpolation step is highly dependent on the domain and the type of vertical coordinate of the target RCM. Here we assume that such an interpolation module exists with the RCM considered. In the case of the COSMO-CLM model (Rockel et al., 2008; Sørland et al., 2021), the procedure is able to handle two height-based terrain-following hybrid coordinates (Schär et al., 2002; Baldauf et al., 2011). Boundary and initial condition files typically contain much more variables than those modified by the PGW approach. These variables (e.g. soil type, liquid and solid water species, aerosol concentrations) are left to the interpolation procedure of the RCM, which anyway must be able to provide some information at the lateral boundaries, irrespective of whether this information is provided by the driving model or not. For instance, the concentrations of condensed water species are often not available, and if present, one needs to keep in mind that they strongly depend upon the microphysics scheme used.

For the PGW simulation, the ERA fields are modified by the climate-change deltas $\Delta$. This step requires the interpolation of the respective fields from the GCM to the ERA computational mesh. The procedure is complicated because of the pressure adjustment that is required (see below).

Alternatively, one could add the deltas directly to the $CTRL_{LBC}$. However, it is more consistent to add these to $CTRL_{ERA}$, in order to ensure that the same interpolation procedures are used for both RCM simulations. Note in particular that the interpolation procedure commonly used to generate the lateral and lower boundary conditions of RCM simulations are complex and far beyond a pure interpolation. For instance they may invoke nonlinear heuristic procedures to account for differences in topographical height. In addition, as all RCMs are equipped to using reanalyses, the ERA file format is a useful interface between our PGW code and the RCM model.

Greenhouse gas (GHG) concentrations can be raised in the PGW simulation consistent with those imposed in the GCM. In practice, this procedure depends upon the RCM considered. It is important to note, however, that the prime effects of changes in GHG concentrations happen in the GCM, where they drive the large-scale warming and moistening. This signal reaches the RCM via the lateral boundaries, and this is more important than local radiative effects in the RCM (e.g. Seneviratne et al., 2002). Nevertheless we recommend adjusting the forcing consistent with the driving GCM, also as the $CO_2$ concentration may affect evapotranspiration, depending upon the land-surface model considered (e.g. Schwingshackl et al., 2019). The situation is similar regarding changes in aerosol concentrations. While representing changes in regional aerosol concentrations is important (e.g. Boé et al., 2020), we are not aware of PGW simulations that fully represent this effect.

## 2.4 Reconstruction of CTRL ERA geopotential

The standard ERA5 reanalysis files contain all relevant three-dimensional fields, but neither the pressure nor the geopotential height field. Rather the surface pressure $p_{sfc}$ is provided. It determines the pressure of all computational levels through the definition of the hybrid vertical coordinate as

$$p_{k+1/2} = a_{k+1/2} + p_{sfc}b_{k+1/2}. \tag{4}$$

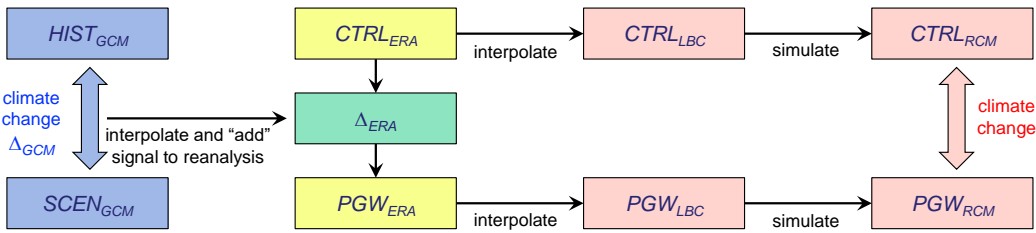

**Figure 2.** Workflow for PGW simulations. The strategy consists of imposing large-scale changes in thermodynamic structure and circulation on a historical RCM simulation, by modifying the lateral and lower boundary conditions correspondingly. The climate-change signal is taken from GCM simulations. Here CTRL and PGW represent the historical and pseudo-global-warming simulations, driven by historical and modulated reanalysis, respectively. The subscripts denote the underlying computational mesh, represented by the GCM, the reanalysis ERA, the regional climate model RCM, and the lateral and lower boundary conditions LBC driving the RCM. Note that the LBC and RCM meshes are typically identical.

Here subscripts $k$ and $k+1/2$ denote the layer centers and interfaces, respectively (numbered from top to bottom), and the coefficients $a$ and $b$ define the hybrid coordinate. As $p_{sfc}$ denotes the surface pressure at the height of the ERA topography, it cannot directly be merged with the $\Delta p_{sfc}$ from the GCM, as the latter is located at a different height. To resolve this problem, $\Delta\phi$ at the GCM pressure level $p_{ref}$ is used instead. In the simplest case, one chooses one single pressure level $p_{ref}$ located above the topography for the whole RCM domain.

The reconstruction of the geopotential on the computational mesh of the ERA reanalysis uses the hydrostatic equation in pressure coordinates

$$\frac{\partial \phi}{\partial p} = -\frac{RT_v}{p}. \tag{5}$$

In numerical terms, this means integrating from the surface ($k = K + 1/2$) to the top using

$$\phi_{k-1/2} = \phi_{k+1/2} - (\ln p_{k-1/2} - \ln p_{k+1/2})\, RT_{v,k} \quad \text{for} \quad k = K, K-1, ... \tag{6}$$

with the lower boundary condition $\phi_{K+1/2} = \phi_{sfc}$. Here $T_v = f(T, q_v)$ or $T_v = f(T, RH, p)$ is known from the definition of the virtual temperature. The geopotential at the reference level, i.e. $\phi_{ref} = \phi(p_{ref})$ is derived using linear interpolation in $\ln(p)$. This yields

$$\phi_{ref} = \phi_{k^*+1/2} - (\ln p_{ref} - \ln p_{k^*+1/2})\, RT_{v,k^*} \tag{7}$$

where $k^* + 1/2$ is the grid point immediately below the $p_{ref}$ level, i.e. $p_{k^*+1/2} \geq p_{ref} \geq p_{k^*-1/2}$. The reference geopotential $\phi_{ref}(x,y)$ will later be combined with the respective changes $\Delta\phi_{ref}(x,y)$ from the GCM simulation.

### 2.5 Interpolation of PGW changes in time

Regional climate models operate with a specified boundary update frequency of typically 1 to 6 hours. To run a PGW simulation, we require the $\Delta$ signal at the same frequency, accounting for the mean seasonal cycle of the signal. We apply two

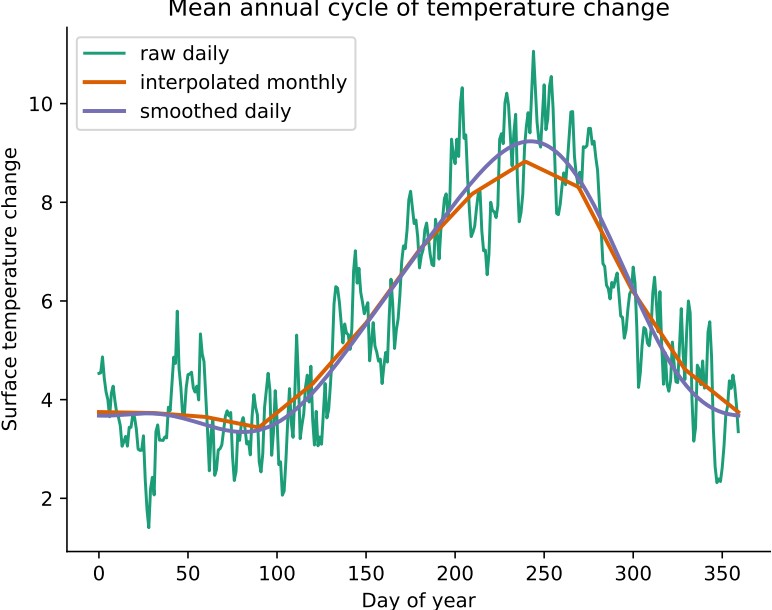

**Figure 3.** Mean annual cycle of the PGW climate-change signal at a specific grid point in Europe for different methods of constructing the annual cycle from GCM data. The green line shows the raw signal from two 30-yr periods at daily resolution. It is strongly influenced by daily and interannual variability. The orange and blue line show different methods of constructing a smooth and continuous annual cycle. The orange line shows the result of linear interpolation between monthly mean values. The blue line is the result of smoothing the daily values using spectral filtering. Both filtering methods are supported by PGW4ERA5.

different methodologies: First, if the computations are based on monthly mean data, the continuous function is computed by linear interpolation (Figure 3, orange line). Second, when using input with daily frequency, the 30-year mean annual cycle (Figure 3, green line) is subjected to the spectral filtering algorithm described in Bosshard et al. (2011), which returns a smoothed annual cycle with daily frequency (Figure 3, blue line). Linear interpolation is subsequently used between the daily values to obtain a continuous function for the annual cycle of $\Delta$ at the boundary update frequency of the RCM.

## 2.6 Application of PGW changes to ERA fields

For each datum in the lateral boundary forcing fields, we add the $\Delta_{\mathrm{GCM}}$ to the ERA reanalysis (see Fig. 2). To this end, $\Delta$ needs to be spatially interpolated from the GCM to the ERA grid, i.e., $\Delta_{\mathrm{ERA}}$. In the horizontal, the changes are bilinearly interpolated. For complicated geographical coordinate systems of the input data set, the xESMF Python package is available (Zhuang et al., 2020). For the three-dimensional variables $\Delta T$, $\Delta u$, $\Delta v$ and $\Delta RH$, we also require an interpolation in the vertical from the GCM pressure levels to the ERA hybrid levels, which again uses linear interpolation in $\ln p$. Ultimately, for three-dimensional

ERA fields $\chi$, the $\Delta$ are applied as

$$\chi'(x,y,p) = \chi(x,y,p) + \Delta(x,y,p) \quad \text{for} \quad \chi = T, u, v, RH. \tag{8}$$

Here $\chi$ and $\chi'$ denote the ERA and PGW variables, respectively.

The pressure values for the ERA hybrid vertical levels depend upon the surface pressure $p_{\text{sfc}}$. When doing the vertical interpolation, we make the assumption that the surface pressure does not change, i.e. $p'_{\text{sfc}} \approx p_{\text{sfc}}$. This is an approximation, but tests demonstrated that relaxing this assumption leads only to minimal changes (see Subsection 2.9.1).

The application of (8) to the two-dimensional surface fields is straightforward. Similarly, the $\Delta$ is also imposed to the geopotential at the reference pressure level $p_{ref}$ as

$$\phi'_{ref} := \phi_{ref} + \Delta\phi_{ref} \tag{9}$$

where $\phi_{ref}$ is defined by (7).

## 2.7 Surface Fields

Over land, the temperature change of the soil levels is derived based on $\Delta T_{\text{sfc}}$ following

$$\Delta T(z) = \overline{\Delta T_{\text{sfc}}} + e^{z/2.8\,m}(\Delta T_{\text{sfc}} - \overline{\Delta T_{\text{sfc}}}), \tag{10}$$

where $\Delta T(z)$ is the soil temperature change at depth $z$ [m] and $\overline{\Delta T_{\text{sfc}}}$ is the annual mean of $\Delta T_{\text{sfc}}$ (i.e., representing the climatological deep soil temperature change). The constant $2.8\,m$ is the penetration depth of the annual cycle for an average thermal conductivity of the soil. A similar procedure is applied when interpolating the driving fields to the mesh of the COSMO models.

Over sea, RCMs typically use prescribed sea surface temperature which is here modified based on $\Delta SST$. The difficult aspect is that $\Delta SST$ is defined on the ocean model grid of the GCM which typically has a complex grid geometry and is undefined over land. The interpolation of $\Delta SST$ to the ERA grid thus requires special attention. The problem is illustrated in Figure 4. Using naive bi-linear interpolation, the missing values (NaN) from land regions are propagated into the ocean $\Delta SST$ field and coastlines are missed (Figure 4, middle panel). A more sophisticated interpolation routine is used instead which ignores the contribution from NaN values. This is done by removing all grid points of the GCM grid over land and re-casting it to an unstructured grid. A Gaussian kernel-based interpolation method (Maz'yai and Schmidt, 1996) is used to interpolate the $\Delta SST$ field onto the ERA ocean grid points. With this approach, a $\Delta SST$ value can be obtained for all water grid points in the ERA data set, and no information is lost at the coastlines (Figure 4, lower panel). A user-defined kernel cut-off value (i.e., the maximum distance across which $\Delta SST$ values are interpolated) can be set to avoid that water basins lacking GCM information get a far-lying $\Delta SST$ value from a remote ocean basin. Instead, for these cases, the method falls back to the $\Delta T_{\text{sfc}}$ to derive the local water surface temperature change. However, as the method operates on $\Delta SST$, completely unrealistic values (that would occur when operating on $SST$ instead) do not occur, even with significant extrapolation.

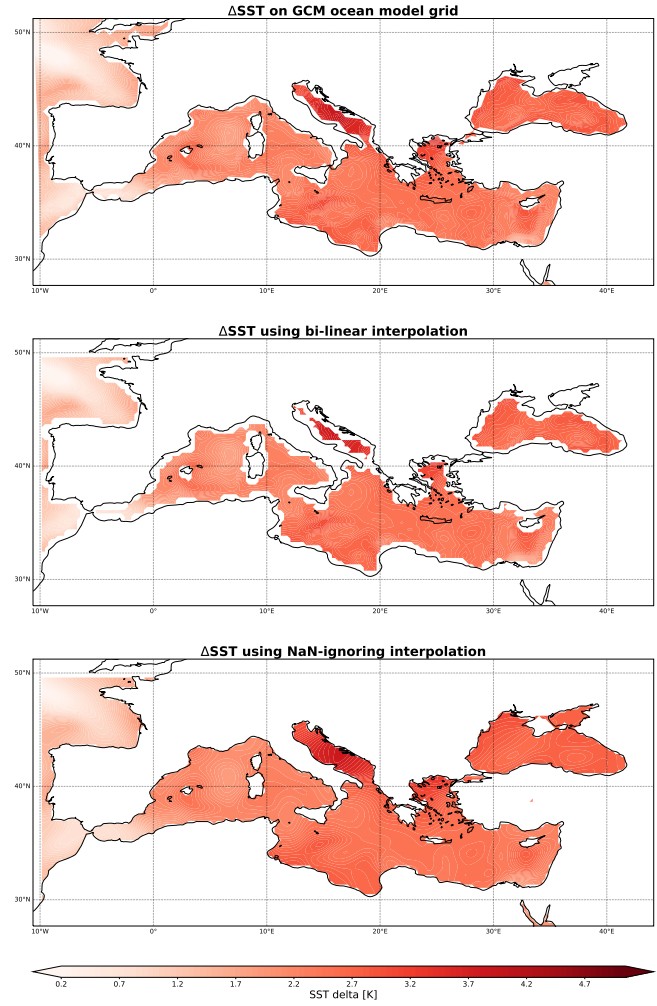

**Figure 4.** Monthly mean sea surface temperature climate delta ($\Delta SST$) during January obtained from a GCM. (upper panel) $\Delta SST$ is shown on the native grid of the GCM ocean model. (middle panel) $\Delta SST$ after naive bilinear interpolation onto the reanalysis grid. Information is lost around the coastlines. (lower panel) $\Delta SST$ after NaN-ignoring interpolation onto the reanalysis grid. Hereby, additional detail near the reanalysis coastlines is achieved through extrapolation of the original $\Delta SST$. Further information about the data and interpolation used is given in Table A1.

In locations with sea ice, the surface temperature of ice is changed according to $\Delta T_{\mathrm{sfc}}$ instead of $\Delta SST$ assuming that the temperature change at the ice surface is independent of the SST change below the ice. Changes in the sea ice fraction between $CTRL$ and $PGW$ are not considered in the current version of the code.

## 2.8 Pressure adjustment

The most demanding task of the PGW procedure is the pressure adjustment. There are two factors to be considered. First, as the troposphere warms, the air will expand, and tropospheric pressure levels are lifted correspondingly. This effect requires a pressure adjustment in the PGW approach (Schär et al., 1996). The magnitude of this effect can be estimated from the hydrostatic relation (5). For further consideration, let's consider the 500 hPa surface. If the air below is uniformly warmed by $\Delta T$, the altitude of the 500 hPa surface will be lifted according to

$$\frac{\Delta z}{\Delta T} = \frac{R}{g} \ln \frac{1000 \text{hPa}}{500 \text{hPa}} \approx 20 \text{m/K} \tag{11}$$

Thus, for each degree of warming, the 500 hPa surface is lifted by about 20 m.

Second, climate change is associated with circulation and pressure changes. In our PGW approach, these changes are entailed in $\Delta \phi_{ref} = \Delta \phi(p_{ref})$. These changes include geographical variations which must also be accounted for.

In pressure coordinates the pressure adjustment is rather straightforward. However, in hybrid (sigma/pressure) vertical coordinates as used by ERA, adjusting the pressure consistently is not straightforward. In particular, the pressure of the coordinate levels depends upon the surface pressure. The unknown in this process is the change in surface pressure $\Delta p_{\text{sfc}}$. An iteration is required with the goal that the resulting change in geopotential height $\Delta \phi_{ref}$ at the reference pressure level $p_{ref}$ agrees with that provided by the GCM.

We start the iteration with $p'_{\text{sfc}}{}^{n=0} := p_{\text{sfc}}$, where $n$ denotes the iteration parameter. The iteration is conducted for each grid column, and each iteration step involves the following computations:

- Step 1: Computation of $q_v$ from temperature and relative humidity in ERA. This is needed as our procedure uses $\Delta RH$ rather than $\Delta q_v$ (see Subsection 4.3). This is followed by the computation of the virtual temperature $T_v$ which appears in the hydrostatic relation.

- Step 2: Reconstruction of the geopotential using the same procedure as described in Subsection 2.4 above. In essence, this is the vertical integration of the hydrostatic equation expressed by (6).

- Step 3: Computation of $\phi'_{ref}{}^n$ at the reference pressure level $p_{ref}$ using (7). Note that this requires the computation of $k^*$, an integer that may change with the iteration.

We may write the iteration step of the pressure adjustment as

$$\phi'_{ref}{}^n = f\left(p'_{\text{sfc}}{}^n\right). \tag{12}$$

There are different ways how to advance the iteration. Simple scaling suggests to proceed with

$$p'_{\text{sfc}}{}^{n+1} - p'_{\text{sfc}}{}^n = -\alpha \frac{p'_{\text{sfc}}{}^n}{RT'_{\text{sfc}}} \left(\phi'_{ref}{}^n - \phi'_{ref}\right) \tag{13}$$

with the target geopotential $\phi'_{ref}$ (obtained from Equation (9)) and a proportionality factor $\alpha \leq 1$. Using $\alpha = 0.95$ and $\phi'_{ref}{}^n - \phi'_{ref} \leq 0.15 \, \mathrm{m^2 \, s^{-2}}$ as convergence condition, the iteration usually requires less than 10 steps to converge. Upon completion of this iteration, the PGW-shifted ERA reanalysis may be used for the computation of the lateral boundary conditions and the execution of the PGW simulation (Fig. 2).

## 2.9 Refined pressure adjustment

### 2.9.1 Refined vertical interpolation

In Subsection 2.6 we have simplified the interpolation in the vertical when transforming $\Delta_{GCM}$ to $\Delta_{ERA}$ by assuming that changes in pressure are small. In principle, one can relax this assumption by including the application of the $\Delta$ (see Subsection 2.6 and (7)) into the iteration loop discussed in Subsection 2.8. We have tested this procedure but did not find significant improvements. Indeed, likely other aspects of the procedure (such as using monthly mean changes, or the vertical resolution of the climate deltas) are more important than the details of the vertical interpolation. Moreover, in the lower troposphere the effect of the pressure adjustment is small according to (11).

### 2.9.2 Use of multiple geopotential height levels

In the version of the code described above, we use one specific $p_{ref}$ level to account for the changes in geopotential. This level must be chosen at a height which is above all topography in the computational domain. If there is high topography within the domain, one might be forced to use an elevated level at the 500 hPa level or even higher. This implies that the vertical integration in step 2 of the iteration (see Subsection 2.8) must cover a deeper layer and might suffer from a larger error (see below). For this reason, the code enables the use of multiple reference levels. In essence, the lowermost pressure level located above the topography in a specific ERA grid column is then used.

The pressure adjustment is affected by the number of pressure levels of the climate deltas. For instance, when using AMON data, there are 19 pressure levels in total, thereof only 6 layers between 1000 and 500 hPa. This has implications on the accuracy of the pressure adjustment via the integration of the geopotential. While in the GCM $\Delta\phi_{ref}$ in (9) is integrated on the native vertical grid of the GCM (using $T$ and $q_v$ on all GCM levels), the integration of $\phi'_{ref}$ used in (13) is an approximation and depends on the vertical resolution of $\Delta T$ and $\Delta q_v$ below $p_{ref}$. An illustration of this is shown in Fig. 5 for both AMON and EMON data on a marine trade-wind domain. When using the low-resolution AMON data, the poorly represented jump in $\Delta T$ and $\Delta q_v$ across the trade-wind inversion leads to errors in the integration of $\phi'_{ref}$ and in the adjusted surface pressure. Such uncertainties are particularly important in the tropics, where small horizontal gradients in geopotential can drive significant circulations. The problem is further illustrated in Fig. 6, showing the deviation in geopotential height at different pressure levels between two PGW setups with EMON and AMON climate deltas. The pressure adjustment is done with one reference pressure level at 500 hPa. Consequently, the deviation (EMON-AMON) is very small at 700 hPa but increases towards the surface and towards the model top. The error resulting from the trade-wind inversion is well visible in the left panel.

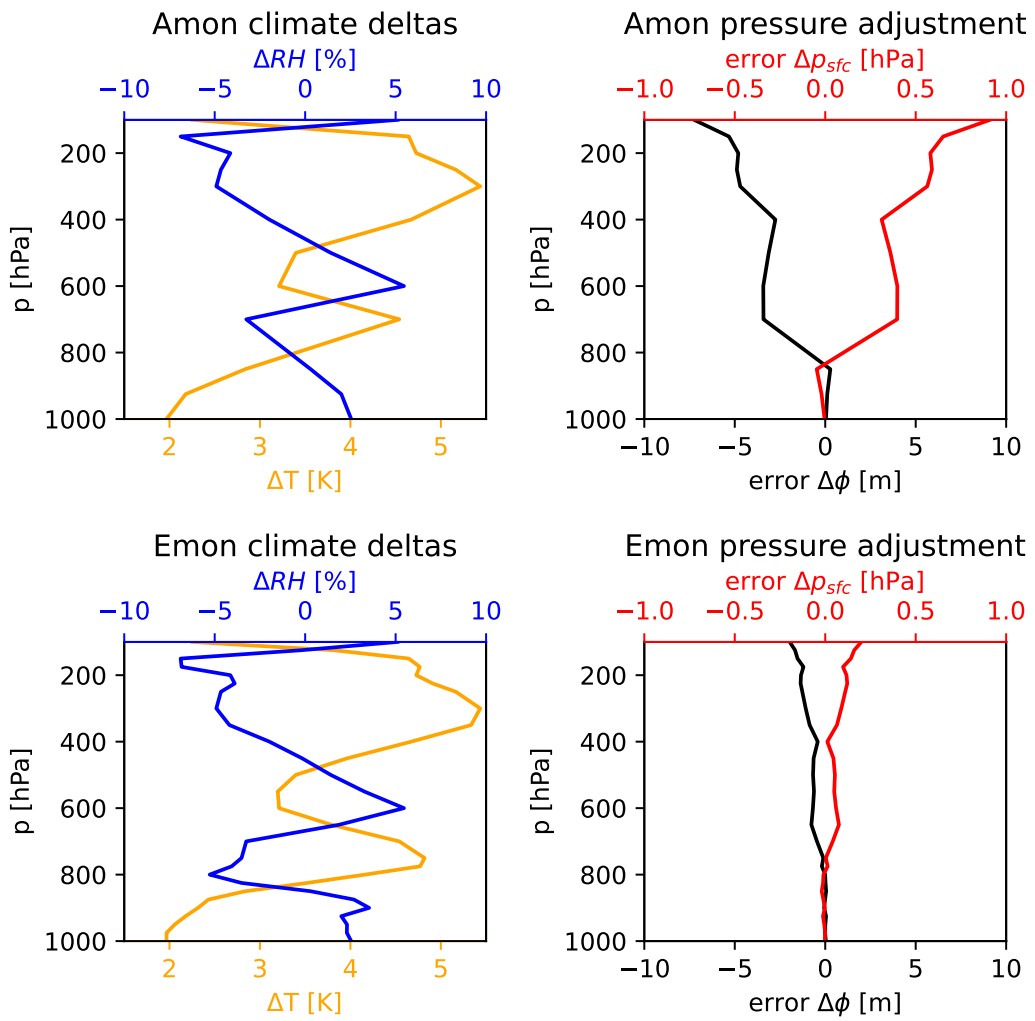

**Figure 5.** Illustration of the pressure adjustment using AMON (top) and EMON (bottom) pressure level data with 19 and 27 pressure levels, respectively. The analysis is done for a 2°x2° box over the subtropical Southern Atlantic at a particular point in time. Left panels show $\Delta T$ and $\Delta RH$, right panels show the integration error in $\Delta \phi$ when integrated from the surface (sfc) to pressure $p$, and the error in $\Delta p_{\text{sfc}}$ resulting from the pressure adjustment for a given reference pressure level $p_{ref}$ (y-axis). The adjustment based on AMON data has large uncertainties (error in $\Delta p_{\text{sfc}}$) for $p_{ref} < 850$ hPa, due to the pronounced trade wind inversion. Further information about the data used and the error computation is given in Table A1.

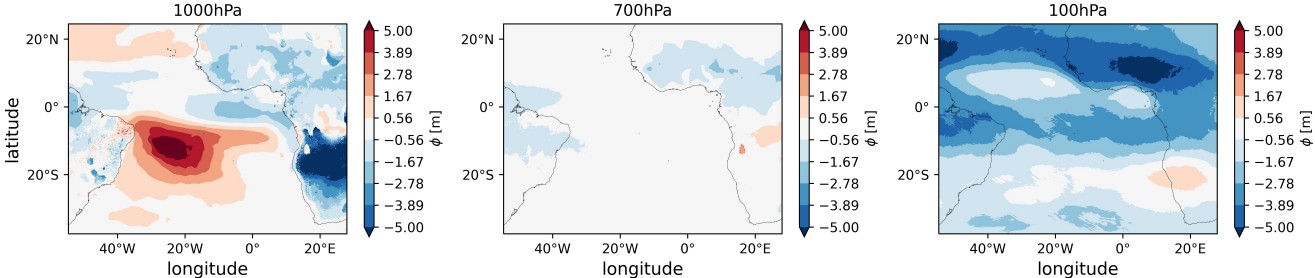

**Figure 6.** Difference in the geopotential height [m] between PGW boundary conditions derived using EMON and AMON data, respectively, at 1000 hPa (left), 700 hPa (center), and 100 hPa (right) at one specific time. In both cases, the pressure adjustment is done with one fixed reference pressure level at 500 hPa. Further information about the data used is given in Table A1.

## 3 Comparison with Standard Dynamic Downscaling

The key question is whether the simplifying assumptions made with the PGW methodology are sufficiently valid. To test this question we present a detailed intercomparison of climate change scenarios derived from both the conventional and PGW methodologies. In the intercomparison, the conventional methodology considers multi-decade-long SCEN and HIST simulations for recent and end-of-century periods driven by a GCM, while the PGW scenario uses the same large-scale forcing but applies it to modify a CTRL simulation. Further details about the simulations, such as information on emission scenarios, the time periods considered, and the driving simulations, can be found in Table A1.

In Figure 7, we show a direct comparison of transient RCM simulations with corresponding PGW simulations. We aimed to assess how similar to a transient simulation a PGW simulation can be. Therefore, to derive the PGW simulation, we used the transient simulation to provide $\Delta$ with (2). We also used the historical period (1971-2000) from the transient simulation as basis for the PGW ($CTRL$ in Equation 1). This means that in Figure 7, the future simulation differs (GCM-driven vs. PGW) but the historical simulation is identical in both cases. The details on the models used can be found in Table A1.

We observe that the temperature projections in the transient simulations and the PGW simulations are virtually identical (Figure 7a-b). Also, the PGW approach leads to similar projections for precipitation (Figure 7c-d) and mean wind (Figure 7e-f). The eddy kinetic energy (EKE, defined as $0.5((u-\overline{u})^2+(v-\overline{v})^2)$ and computed as in Brogli et al. (2019b)), a proxy for cyclone activity, clearly decreases in the GCM-driven simulations (Figure 7h), while in the PGW, EKE does not substantially change (Figure 7g). As discussed earlier, this is the expected behaviour, since a historical simulation is merely modified in the PGW approach and the sequence of cyclones entering the domain does not change. More specifically, the climate change signal exhibits polar amplification and baroclinicity, consistent with a reduction in eddy-kinetic energy (Figure 7h).

Since in the majority of PGW use cases employ a reanalysis-driven historical simulation, we next compare a reanalysis-based PGW simulation to a GCM-based PGW simulation and to standard downscaling. Figure 8 shows changes in precipitation statistics in these three types of simulations (three columns). Independently of the choice of the downscaling strategy, the projected changes in precipitation are extremely similar and show a pattern that is consistent with previous analyses of large

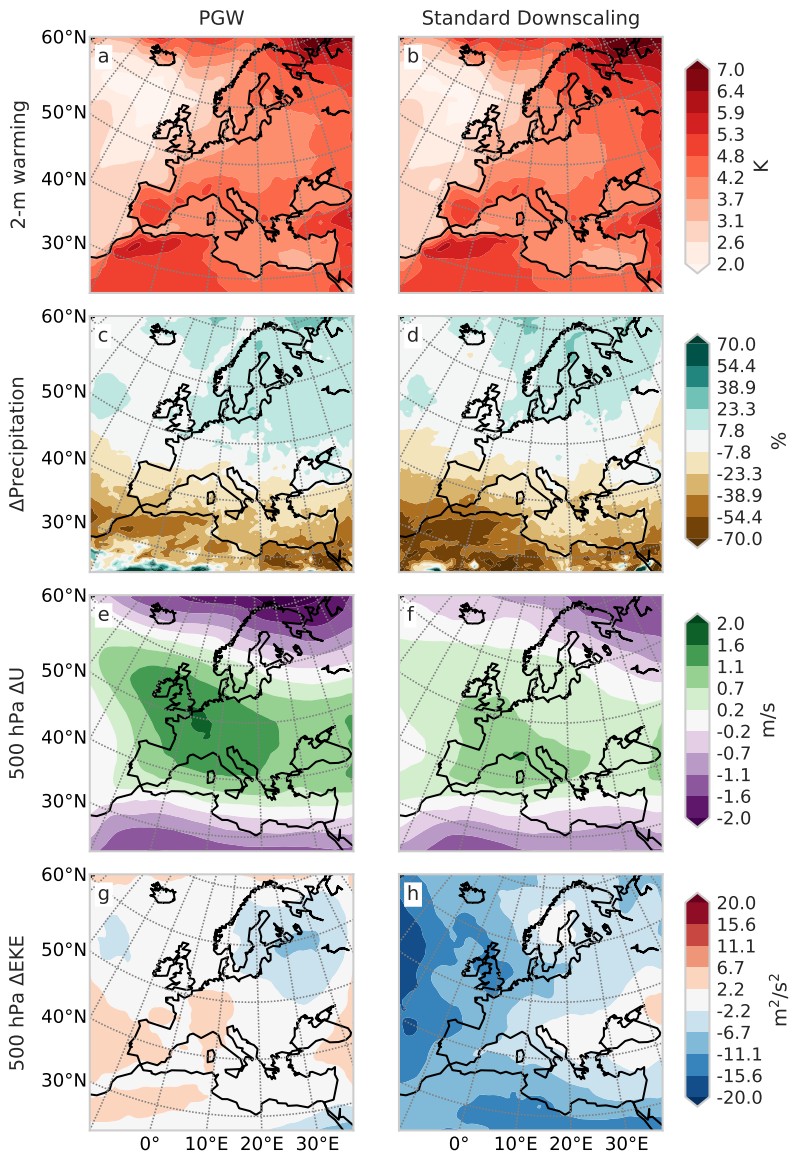

**Figure 7.** Comparison of annual-mean climate-change projections derived from (left) PGW and (right) standard downscaling methodologies for the time periods 2070-2099 versus 1971-2000 using an RCP8.5 emission scenario. (a-b) 2-m temperature change, (c-d) mean precipitation change, (e-f) mid-tropopheric zonal wind change, (g-h) mid-tropospheric change in eddy kinetic energy (calculated as $0.5((u-\overline{u})^2 + (v-\overline{v})^2)$ with $(u,v)$ as the instantaneous value of the wind and $(\overline{u}, \overline{v})$ the 30-yr mean thereof). Details on the simulation set up can be found in Table A1.

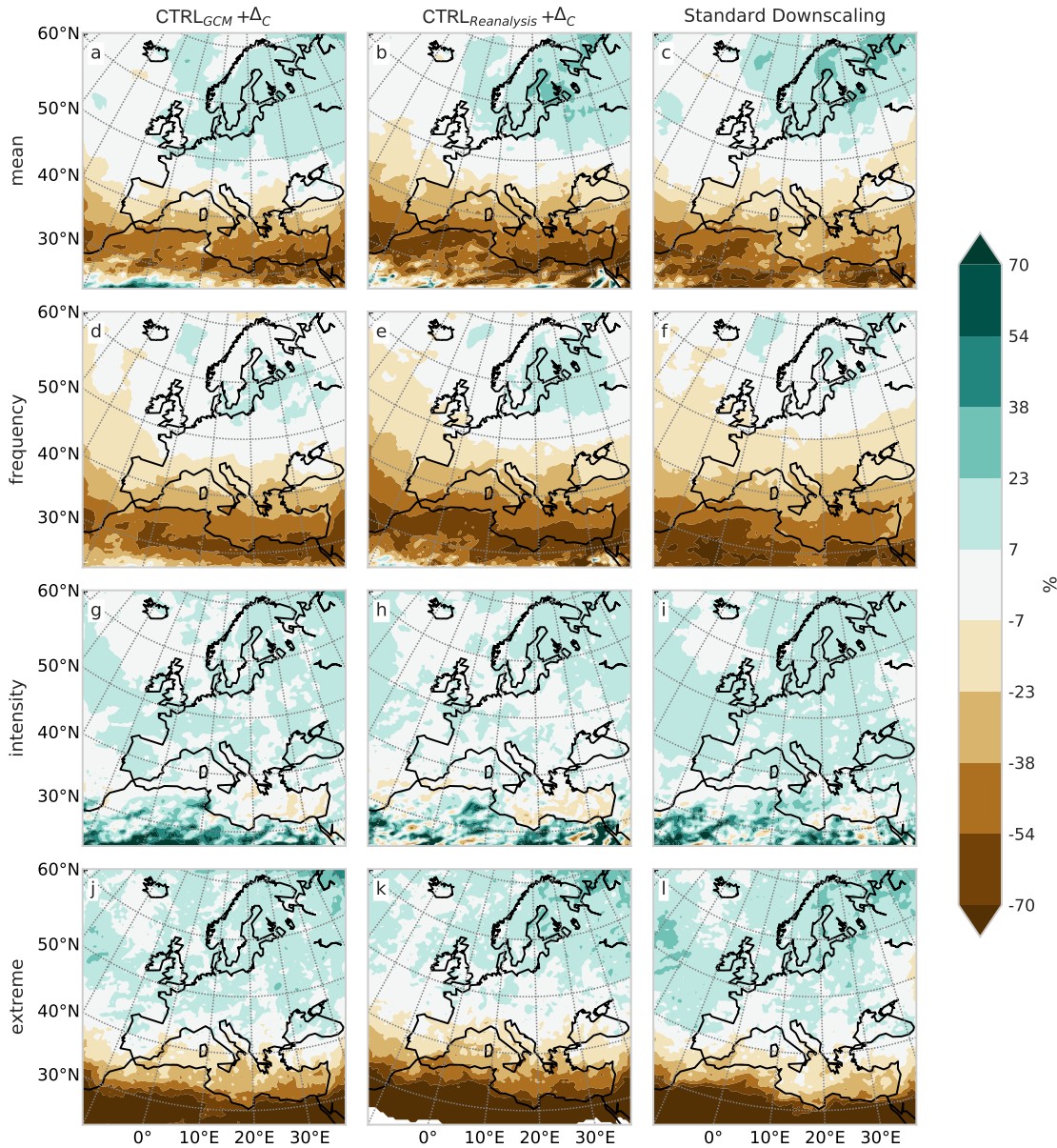

**Figure 8.** Similar to Fig. 7 but for annual-mean changes in (a-c) mean precipitation, (d-f) precipitation frequency (days with precipitation $\geq 1$ mm), (g-i) precipitation intensity (precipitation amount on days with $\geq 1$ mm), and (j-l) extreme precipitation (99th percentile of all days), for three climate change projections. The first column (a,d,g,j) shows a GCM-driven $CTRL$ simulation modified using the PGW approach. The second column (b,e,h,k) shows a ERA-Interim driven $CTRL$ simulation modified using the PGW approach. The third column (c,f,i,l) shows the standard GCM-driven transient climate simulation. Details on the models used are in Table A1.

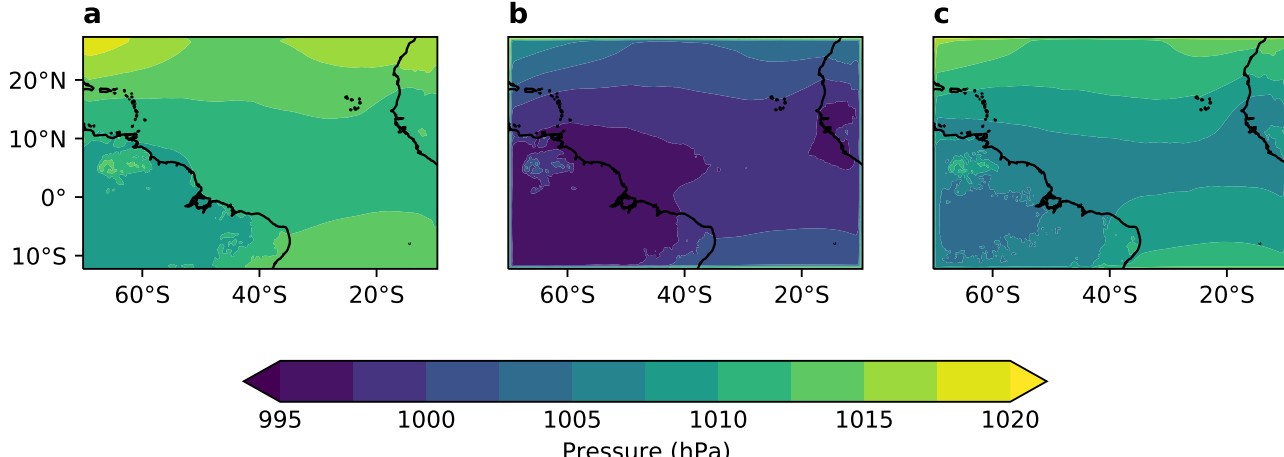

**Figure 9.** Mean sea level pressure in PGW simulations and the effect of maintaining the hydrostatic balance. (a) mean sea-level pressure in the reanalysis-driven CTRL simulation of the month November 2004. (c) shows the PGW simulation where the full PGW procedure described in Section 2 was followed, including the pressure adjustment. (b) as (c), but using a simplified procedure where the pressure at the lateral bondaries was left unchanged. The computational domain covers the tropical Atlantic. Note the significant differences between (b) and (c), highlighting the importance of the pressure adjustment. Technical simulation details are in Table A1.

simulation ensembles (Rajczak and Schär, 2017). This is not only true for mean precipitation changes, but also for changes in the intensity and frequency and extreme indices. The differences between the two PGW versions (columns 1 and 2) are entirely due to the use of another historical simulation, while the $\Delta$ used in both PGW simulations shown in Figure 8 are identical. Although being similar, some subtle differences in projected precipitation changes are visible. It is hard to tell which of the scenarios should be closest to reality. One the one hand, the reanalysis-driven historical simulation has the smallest bias. On the other hand the standard downscaling approach most consistently accounts for large-scale climate changes.

## 4  Sensitivity Tests

During the development of our PGW simulations we performed multiple sensitivity tests related to the design of the workflow which will be presented in this section.

### 4.1  Hydrostatic Balance

We maintain the hydrostatic balance in our PGW simulations by adjusting the pressure in each boundary condition field according to Section 2.8. Figure 9 demonstrates that doing so matters for the simulation result. More specifically, we see that in

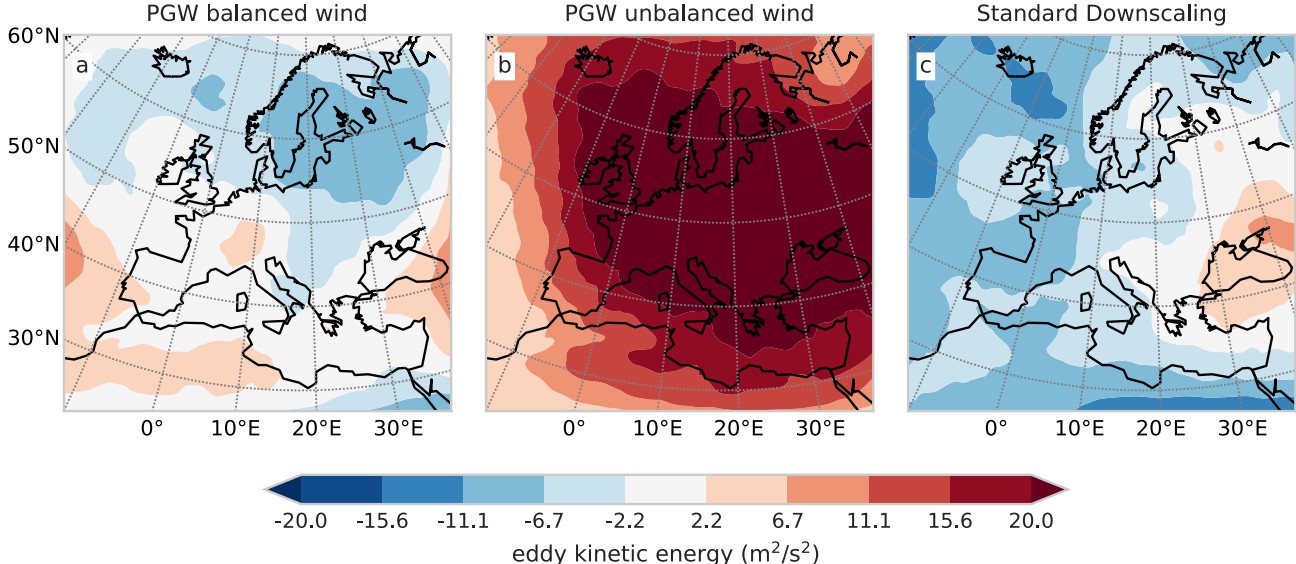

**Figure 10.** Effect of the thermal wind balance on changes in 500 hPa eddy kinetic energy (EKE) in PGW simulations. (a) PGW simulation where both changes in temperature and wind were made at the lateral boundaries and therefore the thermal wind balance was maintained. (b) PGW simulation where the temperature at the lateral boundaries was changed without corresponding changes in wind. (c) EKE changes from dynamically downscaling the same GCM. Details on the simulations in Table A1.

a historical simulation for November 2004 over the tropical Atlantic the mean sea level pressure is $\sim 1010$ hPa. If we raise the temperature in the same simulation and use the same pressure at the lateral boundaries, the mean sea level pressure drops by

10-15 hPa within the domain, which is no longer realistic. Note that this implies a discontinuity at the lateral boundaries, which is visible by close inspection in Figure 9b. By readjusting the pressure, the mean sea level pressure values remain reasonable even when the temperature is raised (Figure 9c).

## 4.2 Thermal Wind Balance

Figure 10 presents the effects of prescribing wind changes consistent with the thermal wind balance at the lateral boundaries

of PGW simulations. We compare a simulation that includes the full PGW methodology as described in Section 2 against one where the 3D temperature changes were applied, but not the corresponding changes in mean wind. We compare the two simulations in terms of EKE. It is evident that the response deteriorates when the thermal wind balance at the lateral boundaries is violated by making temperature changes without corresponding wind changes (Figure 10b). This increase in EKE can physically be interpreted as resulting from a geostrophic adjustment process. Also, when not balancing the wind

changes, the resulting strong EKE change has the opposite sign of what is seen in dynamic downscaling (Figure 10c). Note, that wind changes need to be made along with temperature changes as soon as spatial gradients in the temperature change

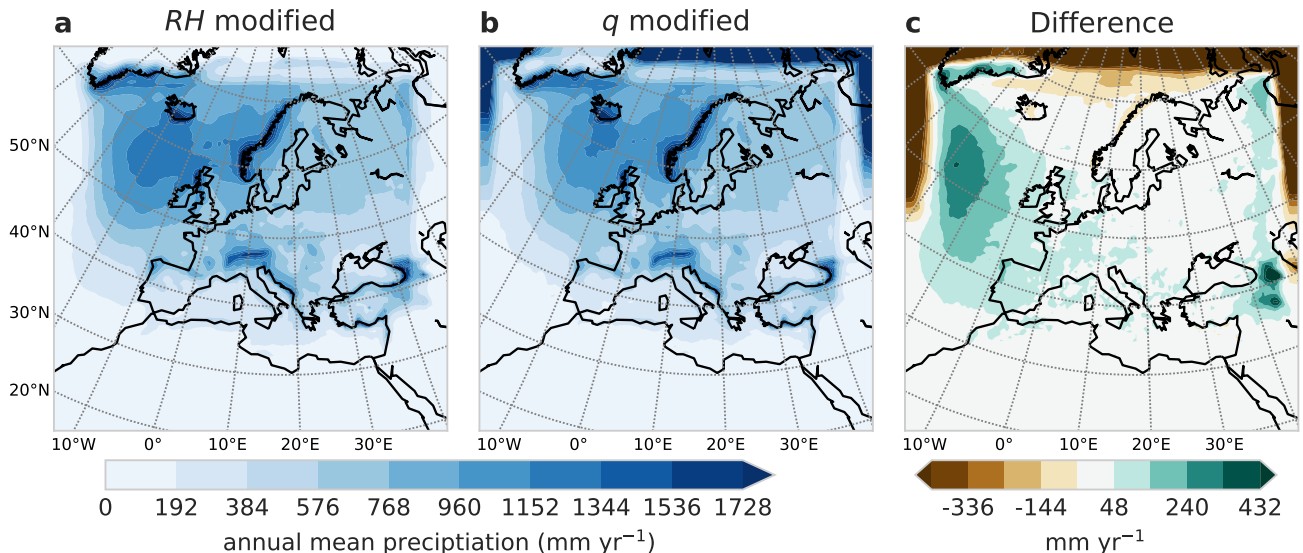

**Figure 11.** Different modifications of humidity in PGW simulations and their effect on annual mean precipitation between 2070-2099. (a) humidity is changed according to the GCM-projected changes in relative humidity. (b) humidity is changed according to projected changes in specific humidity. (c) difference between (a) and (b). The panels show the whole computational domain including the lateral relaxation boundary zone. Details on the models used are given in Table A1.

are prescribed. In idealized PGW settings where a spatially uniform temperature change is imposed, the wind can be left unchanged.

### 4.3 Changes in Humidity

It may be surprising that PGW4ERA5 relies on changes in relative humidity ($RH$) to modify the moisture in the PGW framework, even though most RCMs use specific humidity ($q_v$) as a prognostic and output variable. As shown in Figure 11, a direct modification of the boundary conditions with $q_v$ regionally leads to artificial supersaturation along the model boundary. This supersaturation leads to an unrealistic precipitation band along the model boundary and also affects precipitation in the interior of the domain (Figure 11b,c). This problem is avoided by using $RH$ to modify the moisture (Figure 11a).

How can we explain the difference between these approaches? The $\Delta RH$ for humidity at a certain time step is representative of the climatological change at the time of the year. Thus, virtually the same change is imposed for example during the night (comparably cool and low moisture holding capacity) and during the day (warmer and higher moisture holding capacity), and similar during cold and warm days. During cool periods, it is easy to heavily over-saturate the atmosphere at the lateral boundaries when an absolute change in $q_v$ is imposed. In contrast to $q_v$, $RH$ changes little under climate change, as it intrinsically contains information on the relative saturation of the air. Thus, the changes in $RH$ in the PGW framework can be understood as deviations from the expectations based on the Clausius-Clapeyron relation, and they are typically small (10 % or less).

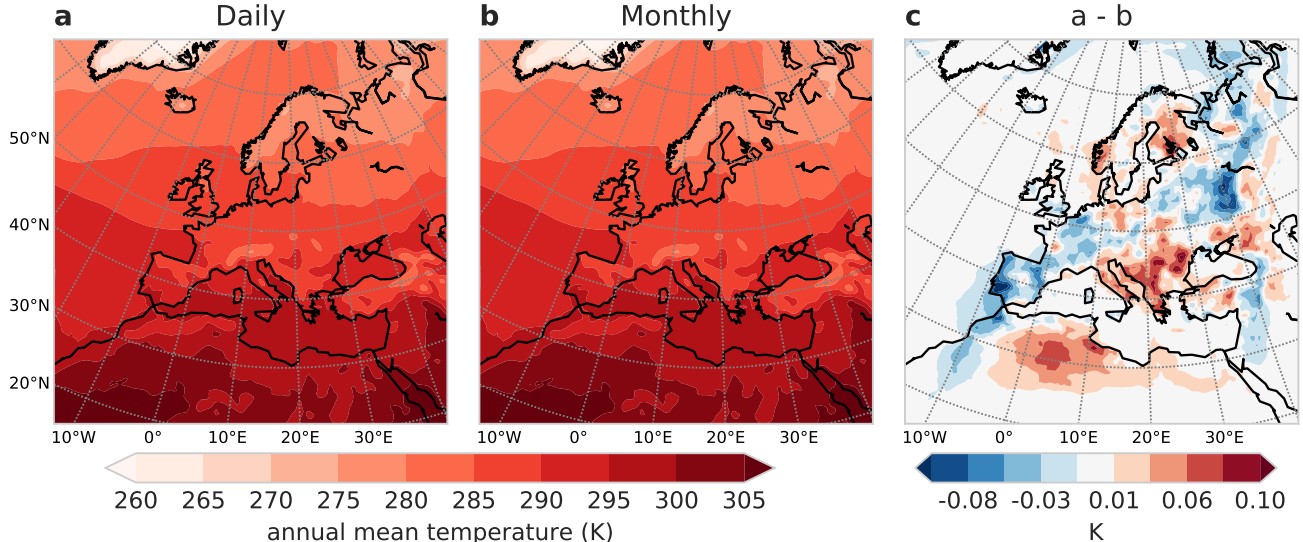

**Figure 12.** Annual mean temperature distribution for 2070-2099 in a PGW simulation depending on the selected temporal resolution of the input data. (a) PGW simulation where smoothed daily changes have been used to modify the lateral boundaries (blue line in Figure 3). (b) PGW simulation where a linear interpolation between monthly mean changes was applied (orange line in Figure 3). (c) difference between (a) and (b).

## 4.4 Temporal resolution of input data

The mean annual cycle of $\Delta$ can be computed based on monthly mean or daily mean input data in PGW4ERA5 (see Section 2.5). Our tests have shown, that the choice of the procedure has only a minor effect on the simulation results on climatological time scales. An example for 30 yr annual mean temperature is shown in Figure 12, where we observe that the mean pattern of future temperature is almost indistinguishable when comparing a PGW simulation based on daily and monthly mean input. The small differences between the approaches ($< 0.1$ K) are likely due to chaotic dynamics and associated sensitivities due to small changes (Figure 12c) . These same results also hold for precipitation changes (not shown).

Based on these tests, using monthly mean input data is the preferable strategy in the majority of PGW applications. That is because it is less data and compute intensive to process the monthly mean values for $SCEN$ and $HIST$. Still, if daily input data (with the same or higher vertical resolution) is readily available it may make sense to use it.

## 5 Conclusions

We have shown that the pseudo-global warming (PGW) approach is an alternative method to provide climate scenarios using regional climate models. If adequately designed, PGW simulations provide plausible future climate change projections that agree well with traditional dynamic downscaling. The methodology uses global model output and accounts for large-scale

changes in temperature and humidity, as well as monthly-mean circulation changes. However, it does not account for changes in large-scale interannual variability (e.g. changes in the frequency of El Nino). In our paper we present detailed intercomparison against the standard GCM-RCM downscaling approach. For most of the impact-oriented fields, such as surface temperature and precipitation changes, as well as precipitation indices relating to heavy and extreme precipitation events, the differences between the two approaches are small and sometimes barely noticeable. As one would expect, there are some minor changes in upper-level synoptic activity, but these do not appear to significantly affect the impact-oriented output parameters.

PGW simulations can be attractive for reducing the computational burden of climate projections, offer flexibility in the design of future projections, and allow future simulations based on reanalysis-driven evaluation runs.

Still, expertise is required to prepare a PGW simulation. With an extended description and evaluation of the methodology, and by providing the PGW4ERA5 software, this work intends to support the future use of PGW simulations. The workflow is generalized where possible and completely written using the widely used Python programming language. The interface to the RCM is provided on the level of the ERA5 reanalysis, i.e. the software provides modified ERA5 reanalysis files.

*Code availability.* The PGW4ERA5 software can be obtained from https://github.com/Potopoles/PGW4ERA5 under the doi https://doi.org/10.5281/zenodo.6627081. The weather and climate model COSMO is free of charge for research applications (for more details see: http://www.cosmo-model.org).

**Appendix A**

**Table A1.** Most figures presented in this article show climate simulations. This table contains details on the set up of the simulations shown in the Figures. The first column indicates the figure for which the information is shown. The second columns shows the panels or parts of the figures for which the information is valid. The description of the simulations is in the third column.

| Figure | Simulation / Panel | Properties |
| --- | --- | --- |
| Figure 1 | (a) | The RCM is COSMO-crCLIM (Leutwyler et al., 2017) at a horizontal resolution of 50 km. The driving data is the ERA-Interim reanalysis (Dee et al., 2011). |
| Figure 1 | (b) | The same RCM as above but using PGW. Initial and boundary conditions from ERA-Interim are modified based on the climate change signal from a transient COSMO-CLM4.8 (Rockel et al., 2008) simulation, dynamically downscaling MPI-ESM-LR (Stevens et al., 2013). The climatic change is representative for the difference between 1971-2000 and 2070-2099 assuming RCP8.5 on a daily timescale. |
| Figure 4 | all panels | $\Delta SST$ is given by the GCM MPI-ESM1-2-HR (Gutjahr et al., 2019) representing the change between 1985-2014, and 2070-2099 following the SSP5-8.5 (Meinshausen et al., 2020) emission scenario and is shown for the month January. The CMIP6 output group OMON (for ocean data) is used. The data is interpolated onto the ERA5 grid in the middle panel using bilinear interpolation and in the lower panel using Gaussian kernel based interpolation with a kernel cut-off at 300 km. |
| Figure 5 | upper panels | ERA5 data on 01.08.2006 00:00 UTC averaged between 10°W-8°W and 10°S-8°S. Similar as for Figure 4, the $\Delta$ are given by the GCM MPI-ESM1-2-HR representing the change between 1985-2014, and 2070-2099 following the SSP5-8.5 emission scenario and interpolated in time to the ERA5 time step. The CMIP6 output group AMON is used. |
| Figure 5 | lower panels | Like upper panels, but the output group EMON is used to compute the $\Delta$. |
| Figure 5 | right panels | The error in $\Delta\phi$ is estimated by comparing $\phi'_{\text{ref}} - \phi_{\text{ref}}$ for a given $p_{ref}$ with the climate delta $\Delta\phi$ at $p_{ref}$, where $\phi_{\text{ref}}$ and $\phi'_{\text{ref}}$ are computed following (7) before and after applying the climate perturbation, respectively. Since the computation is performed over ocean grid points with zero surface elevation in both the GCM and the reanalysis data set, the corresponding surface pressure change $p'_{\text{sfc}}{}^{N} - p_{\text{sfc}}$ (where $p'_{\text{sfc}}{}^{N}$ is computed following (12) and $N$ denotes the last iteration of the pressure adjustment) should be equal to the climate delta $\Delta p_{\text{sfc}}$ which allows to derive the error in $\Delta p_{\text{sfc}}$ as a function of $p_{ref}$. |

| Figure 6 | all panels | Difference between the initial conditions of two PGW simulations conducted with the RCM COSMO on 01.08.2006 00:00 UTC. This means that the ERA5 files are already converted to model boundary conditions by the respective software used in COSMO. One simulation uses $\Delta$ from the CMIP6 output group EMON with high vertical resolution, while the other one uses AMON. Besides this difference, the $\Delta$ are identical to Figure 5. |
|---|---|---|
| Figure 7 | left column (a,c,e,g) | Average of two PGW simulations using the RCM COSMO-CLM4.8. Horizontal resolution is 50 km. For one simulation $\Delta$ is based on MPI-ESM-LR, for the second on HadGEM2-ES (Bellouin et al., 2011). The changes are representative of the emission scenario RCP8.5 and the difference between 1971-2000 and 2070-2099 on a daily timescale. |
| Figure 7 | right column (b,d,f,h) | Average of two transient COSMO-CLM4.8 simulations from 1950-2099 assuming RCP8.5. The simulations are driven by MPI-ESM-LR and HadGEM2-ES. Horizontal resolution is 50 km. The plots show the change between 1971-2000 and 2070-2099. |
| Figure 8 | left column (a,d,g,j) | PGW simulation, where $CTRL$ is COSMO-CLM4.8 downscaling MPI-ESM-LR for 1971-2000. $\Delta$ is also calculated based on a COSMO-CLM simulation downscaling MPI-ESM-LR and represents the 2070-2099 period assuming RCP8.5 on a daily timescale. The horizontal resolution is 50 km. |
| Figure 8 | middle column (b,e,h,k) | Same as left column simulations but $CTRL$ is driven by ERA-Interim. |
| Figure 8 | right column (c,f,i,l) | Same RCM as the rest of the figure but showing a transient simulation driven by MPI-ESM-LR from 1950-2099 assuming RCP8.5. |
| Figure 9 | (a) | Simulation with the RCM COSMO-crCLIM at a horizontal resolution of 12 km. Boundary conditions provided by ERA-Interim. |
| Figure 9 | (b) and (c) | Same simulation as in (a) but PGW-modified with $\Delta$ given by the GCM MPI-ESM1-2-HR (Gutjahr et al., 2019) representing the change between 1985-2014 and 2070-2099 following the SSP5-8.5 (Meinshausen et al., 2020) emission scenario. The input data is given as monthly mean values. |
| Figure 10 | (a) | PGW simulation, where $CTRL$ is COSMO-CLM4.8 downscaling MPI-ESM-LR for 1971-2000. $\Delta$ is also calculated based on a COSMO simulation downscaling MPI-ESM-LR and represents the the difference between 2070-2099 and 1971-2000 assuming RCP8.5 on a daily timescale. The horizontal resolution is 50 km. |
| Figure 10 | (b) | Same as (a) but no changes in the wind have been made at the lateral boundaries. |
| Figure 10 | (c) | Same RCM as the rest of the figure but showing a transient simulation driven by MPI-ESM-LR from 1950-2099. The plot compares the periods 1971-2000 and 2070-2099. |
| Figure 11 | (a) | PGW simulation, where $CTRL$ is COSMO-CLM4.8 downscaling HadGEM2-ES for 1971-2000. $\Delta$ is also calculated based on a COSMO simulation downscaling HadGEM2-ES and represents the difference between 2070-2099 and 1971-2000 assuming RCP8.5 on a daily timescale. The moisture variable modified at the lateral boundaries is $RH$. The horizontal resolution is 50 km. |

| Figure 11 | (b) | Same as (a) but the moisture variable modified at the lateral boundaries is $q_v$. |
| Figure 11 | (c) | Same RCM as the rest of the figure but showing a transient simulation driven by HadGEM2-ES from 1950-2099. The plot compares the periods 1971-2000 and 2070-2099. |
| Figure 12 | (a) | PGW simulation, where $CTRL$ is COSMO-CLM4.8 downscaling HadGEM2-ES for 1971-2000. $\Delta$ is also calculated based on a COSMO simulation downscaling HadGEM2-ES and represents the difference between 2070-2099 and 1971-2000 using daily mean values. RCP8.5 is assumed and the horizontal resolution is 50 km. |
| Figure 12 | (b) | Same as (a), but the difference between 2070-2099 and 1971-2000 has been expressed as monthly mean values. |

*Author contributions.* R.B., C.H. and J.M. wrote the software, C.S. designed the pressure adjustment methodology and drafted Section 2, C.H. and R.B. ran the simulations and prepared most of the figures, R.B., C.S, C.H. and S.L.S. and wrote the article.

*Competing interests.* The authors declare no competing interests.

*Acknowledgements.* The authors want to thank 2 anonymous reviewers for their constructive inputs to the manuscript. We acknowledge PRACE for awarding us access to Piz Daint at Swiss National Supercomputing Center (CSCS, Switzerland). Furthermore, we acknowledge the COSMO, CLM and C2SM communities for developing and maintaining COSMO in climate mode. This project has received funding from the European Union's Horizon 2020 research and innovation programme under grants agreements No 776613 (Project EUCP) and No
820829 (Project CONSTRAIN), and from the Swiss National Science Foundation under number 192133 (Project trCLIM). ERA5 reanalysis data was downloaded from the Copernicus Climate Change Service (C3S) Climate Data Store.

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
