# Peer review of "The pseudo-global-warming (PGW) approach: Methodology, software package PGW4ERA5 v1.1, validation and sensitivity analyses"

_Geoscientific Model Development, 2022_

## Author Response (AR1)

We would like to thank the two reviewers for their very useful and constructive feedback. We think they looked at the manuscript carefully and brought up important points which helped to improve the final version of the manuscript. In the following we list reviewer comments 1 (RC1) and 2 (RC2) and our responses.

**RC1**

**General comments:**

This paper presents a detailed description of the methodology in the preparation of the boundary conditions for PGW simulations, provided in the companion software PGW4ERA5. As the authors have said, the PGW approach offers several benefits, so it will be attractive not only for climatologists but also for the groups who are not familiar with atmospheric dynamics and have interest in impact assessment of future climate change in a certain field. The proposed software and this description paper must support such groups.

This paper basically includes sufficient information as a description paper of PGW4ERA5. This paper is also worthy in that the several specific considerations when preparing boundary conditions for PGW simulations are described based on the authors' knowledge and results of sensitivity experiments. On the other hand, there are somewhat insufficient points in terms of discussing how appropriate it is to create boundary conditions for PGW experiments. Whether those insufficient points should be included or not may be a matter of opinion. However, since this paper is expected to be useful to readers who want to know the specific procedures of the PGW method, and not just a description paper, it is suggested the authors revise the manuscript following the comments below prior to publication.

**Specific comments:**

[1]

The method presented in this paper to create the lateral boundary data for a PGW experiment is one of several options; the procedure in this paper uses ERA data on the original (hybrid) coordinate as a base climate and the changes in geopotential at the reference level obtained from pressure level GCM data. On the other hand, the simplest and easiest option may be the case of using pressure level data both for reanalysis (base climate) and GCM (climate change) data. In this case, one can simply add $\Delta\varphi$ given from the GCM to the reanalysis data without pressure adjustment. The impacts on the RCM (PGW) results of using reanalysis data on pressure level instead of the original ERA-coordinate as a base climate should be mentioned, because many RCM users usually use pressure level data for boundary conditions.

The reviewer is correct regarding the alternate option to using pressure surfaces. In the revised version we will mention this option explicitly. Indeed, using pressure-level data (for the ERA and the PGW-GCM) simplifies the pressure adjustment. However, we prefer to use the raw ERA data on hybrid-pressure level, for the following reasons:

(1) Using hybrid-pressure allows to use the full vertical resolution of the ERA product. This is for instance essential in cases where there are pronounced inversions.
(2) In addition, using pressure level data implies some complications if there is topography near the lateral boundaries.

The boundary conditions are further converted to the RCM coordinate for calculation because the coordinate of a GCM or reanalysis providing boundary conditions usually differs from that of the RCM. At that time, the bias in pressure adjustment generated in the conversion procedure to the RCM coordinate will be more significant in the case of using pressure level data than the original coordinate data. Therefore, it is also important to evaluate the magnitude of the bias and to indicate the authors' opinion on the use of pressure level data for base climate.

The role of potential biases in the pressure adjustment is illustrated in Fig.5 (numbering of revised manuscript) and associated text. We believe the main challenge is the vertical resolution of the GCM data. In our experience, only a limited number of GCMs provides global high-resolution data while the standard model output (Amon, see Section 2.2) has very limited vertical resolution, and the uncertainty is then dominated by the GCM resolution.

[2] Sec. 2.6 (L. 246): How do you determine the convergence of iterations? Please describe the definition of convergence determination.

With a user-defined threshold value in the geopotential deviation, currently set to 0.15 m2 s-2. This was clarified in the text.

[3] L.267-271 and Figure 4

How were "errors in the integration of $\varphi'_{ref}$ and in the adjusted surface pressure" obtained? Please describe more in detail. For example, does it mean that the error in $\Delta\varphi_{ref}$ in Figure 4 is $\Delta\varphi'^{N}_{ref} - \Delta\varphi^{GCM}_{ref}$? Note that, $\Delta\varphi'^{N}_{ref} = \varphi'^{N}_{ref} - \varphi_{ref}$ and $N$ is the number of $n$ when iteration is converged, and $\Delta\varphi^{GCM}_{ref}$ is climate change of $\varphi$ at the reference level obtained from GCM. However, the iteration should be performed until $\Delta\varphi'^{N}_{ref}$ agrees with $\Delta\varphi^{GCM}_{ref}$, as described in L.229-231. Why are there large differences as shown in Figure 4? Alternatively, does Figure 4 show the results of the difference between $\Delta\varphi^{GCM}$ recalculated by Eq. (7) using only AMON/EMON data and $\Delta\varphi$ directly provided in the AMON/EMON data, which is calculated using the native grid data? Either way, a more careful description is needed.

We agree with the reviewer that this was not explained in detail in the original manuscript. We added a detailed description of the error estimation in Appendix table.

[4] Section 4.3

There are other ways to treat humidity changes in PGW methods; for example, there is the idea of not considering the change in RH (as introduced in Sec. 4.5.2 of Adachi and Tomita, (2020)). It is better to mention those other methods. In addition, when the temperature is below 0°C (upper atmospheric level), there are two definitions of RH. It is possible that the definitions may differ between the reanalysis data and a GCM. In such cases, simply summing them is undesirable.

We specified that there are alternative approaches (such as no change in RH) in Section 2.2. However, based on inspection of GCM climate deltas, we believe that the assumption of zero RH change does not hold for large parts of the planet. We also mention the concerns about different definitions of RH below 0°C in the manuscript and how it is computed in PGW4ERA5.

**Technical corrections:**

[1] L.50-54: What is described here is correct, however, this explanation may lead readers to misunderstand that this paper focuses on the preservation of the hydrostatic balance when converting from the driving reanalysis/PGW coordinate (i.e., boundary conditions) to the RCM coordinate, for instance, $HIST_{ERA}$ to $HIST_{LBC}$ or $PGW_{ERA}$ to $PGW_{LBC}$ in Figure 2. In fact, the paper explains how to maintain the hydrostatic balance when adding the climate change Δ in the GCM coordinate system to the base climate (i.e., reanalysis data), although the concept is the same in either case. In other words, the treatment of hydrostatic balance when converting boundary conditions with the driving reanalysis/PGW coordinate to the RCM coordinate depends on a used RCM's internal procedure.

We agree that this can be confusing. We adjusted the text to refer to the coordinates of the boundary conditions rather than the RCM.

[2] L.70: "Figures 1 and 6" --- "Figure 1 and Figures 6e and f, respectively" will be better.

Since the reference to Fig. 6 is overall a bit confusing here, we decided to only refer to Fig. 1.

[3] L.93-95: While it may be difficult to refer to all the studies using the PGW method, it is suggested to cite several pioneering studies. For example, there are studies that investigated changes in precipitation (Sato et al., 2007; Kawase et al., 2009), temperature changes (Adachi et al., 2012), and snow changes (Hara et al., 2008).

- Sato et al., 2007, Journal of Hydrology, https://doi.org/10.1016/j.jhydrol.2006.07.023
- Kawase et al., 2009, JGR, https://doi.org/10.1029/2009JD011803
- Adachi et al., 2012, JAMC, https://doi.org/10.1175/JAMC-D-11-0137.1
- Hara et al., 2008, Hydrological Research Letter, https://doi.org/10.3178/hrl.2.61

We are thankful for these inputs and added the references.

[4] L.115-122: The difference between "Complete GCM output" and "CFDAY data" is not clear. Both data have the same category on a temporal resolution, i.e., daily, and a spatial resolution, i.e., the original/native (GCM) grid.

We agree that this is inconsistent. CFday is an example of "complete GCM output". We adjusted the listing accordingly. Thanks for pointing this out.

[5] L.161-162: Please add a reference related to "nonlinear heuristic procedures", if possible.

Unfortunately, these procedures are not well documented for the model known to the authors. But it is clear that such routines are required for instance in the case of topographic differences between the reanalysis and the model grid.

[6] L.202-204: If my understanding is correct, the description here is not accurate. Δ in the GCM coordinate system is interpolated to the ERA grid to add it to ERA reanalysis, and then the merged data (Δ+ERA) is regridded from the ERA grid to the coordinate of the target RCM.

Yes, of course. This was a remainder from a previous version of the software. Thanks!

[7] L.252: It would be better to add "Eq.(7)" such as "(see Subsection 2.5 and Eq. (7))"

Great point. Thanks.

[8] L.266: Does it mean "on the native vertical grid *of the GCM*"?

Yes.

[9] Caption of Figure 4: What is the $P_s$ in the caption? Does it mean $P_{sfc}$?

We have made it consistent.

[10] L.322: The definition of EKE would be better moved to L.291, where EKE first appears.

Thanks for the careful inspection! We changed this.

**RC2**

The pseudo-global warming (PGW) approach, pioneered by Schar et al (1996), has been widely used in the regional climate modeling community as an alternative to the traditional GCM-based dynamical downscaling strategy. This manuscript presents an overview of the methodology and provides a detailed description of the forcing construction for PGW simulations in the case of the COSMO-CLM regional climate model (RCM) and ERA5 reanalysis. The pressure adjustment is particularly highlighted to maintain the important physical and dynamical balances in large-scale motions, such as the hydrostatic balance, the thermal wind balance, and the geostrophic balance. The methodology is validated by comparing against the standard GCM-RCM simulation and a set of sensitivity tests. Along with the development of a Python-based software package (i.e., PGW4ERA5), this work will greatly facilitate the preparation and implementation of PGW-type simulations and further promote the application of the PGW approach in the regional climate modeling community. I have a few comments for the authors' consideration and clarification.

1. The authors focus on the construction of the three-dimensional fields which are required for deriving the lateral boundary conditions for PGW simulations, but completely ignore the discussion on the construction of lower boundary conditions such as sea

surface temperature, sea ice, sea level pressure etc. As well as the text, the workflow in Figure 2 needs revisions to include the missed information of lower boundary conditions.

We agree with the reviewer that the discussion of the lower boundary conditions was not detailed enough in the original version. We thoroughly revised it and added Subsection 2.6.1 to the manuscript. In this context, we also added the new Figure 4.

2. The proposed procedures for PGW simulations are specifically designed for the COSMO-CLM model, but are generally applicable to other RCMs. Nevertheless, some adjustments may be necessary when the software package is applied to other models because of different initialization and input data processing strategy. For example, in the case of the WRF model surface pressure is computed using the input sea level pressure and geopotential at pressure levels by the standard initialization module, and the computed surface pressure and input temperature fields are then used to reconstruct the geopotential using the hydrostatic equation. In this particular case, perhaps the pressure adjustment is only required for surface pressure.

The pressure adjustment is done only for the surface pressure, but implicitly this affects the distribution throughout the atmosphere as the surface pressure is the basis for the height of hybrid-pressure levels.

The sea level pressure change would need to be computed manually by integrating the hydrostatic equation from the surface elevation to MSL. This could in principle be implemented in the software.

3. As far as I'm aware, outside Europe the 38-pressure-level ERA5 data are most commonly used, instead of the native model level data; the latter is much larger and its downloading is time-consuming.

See our reply to RC1 [1].

4. English editing is needed to correct grammatical errors.

---

## Referee Report (RR1)

Review comments for GMD-2022-167

"The pseudo-global-warming (PGW) approach: Methodology, software package PGW4ERA5 v1.1, validation and sensitivity analyses" submitted to GMD.

**Summary:**

The questions and comments I pointed out in the first review have been appropriately addressed. The manuscript has been adequately improved. I think this paper is suitable for publication in GMD. I will give several suggestions below as minor/technical comments.

**Minor/technical comments:**

[1] L. 113

If the method you mentioned here is the same as one of Misra and Kanamitsu (2004), you should be better to add it to the reference;

*Misra, V., and Kanamitsu, M. (2004). Anomaly nesting: A methodology to downscale seasonal climate simulations from AGCMs. Journal of Climate, 17, 3249–3262.*

[2] L. 114-116

"one has to use some reanalysis ERA, …" might be better like "one has to use some reanalysis such as ERA, …".

*ERA* and *HIST* in "$\Delta$ = ERA – HIST" are long-team mean, whereas *CTRL* and *HIST* in "CTRL = HIST + $\Delta$" are typically 1 to 6 hourly data. Thus, *HIST*s are used in different senses. It would be better to state that ERA and HIST in "$\Delta$ = ERA – HIST" are long-team mean or climate mean. Otherwise, the readers might misunderstand it like CTRL = HIST + $\Delta$ = HIST + (ERA – HIST) = ERA.

[3] Figure 2

This is just a recommendation; it will be better to add $\Delta_{GCM}$ at "climate change" in blue in Figure 2, which will support the reader's understanding.

Caption of Figure2:

"The subscripts denote the underlying computational mesh" --- The computational mesh of LBC and RCM is generally the completely same, isn't it?

[4] L. 235-236

This is also just a suggestion; it would be more clear if you write "…, we add the $\Delta_{GCM}$ to the ERA reanalysis (see Fig. 2). To this end, … from GCM to the ERA grid, i.e., $\Delta_{ERA}$," because $\Delta$ is used as both meanings of $\Delta_{GCM}$ and $\Delta_{ERA}$, which is a bit complicated.

[5]

I suggest swapping Sec. 2.7 and Sec. 2.8, since the explanation in Sec. 2.8 is a continuation of that in Sec. 2.6.

[6] L.290

"Computation of $q_v$ from temperature and relative humidity" --- Does it mean 'temperature and relative humidity, i.e., $\chi'$ in Eq. (8)'?

[7] Caption in Figure 8

"Change" after "in (a-c)…" may not be necessary; that is, "…for annual-mean changes in (a-c) mean precipitation, (d-f) precipitation frequency …".

---

## Author Response (AR2)

**Authors Response to Reviewer**

We want to thank the reviewer for carefully reading the manuscript and the additional inputs. We gladly implemented these inputs as they make the text clearer. We did not swap Sections 2.7 and 2.8 (reviewer comment [5]) as we feel that this would not improve the order since Section 2.9 is a follow-up of Section 2.8.

Also, we would like to thank the editor for his work.